# Extended Spectrum Beta-Lactamase-Resistant Determinants among Carbapenem-Resistant *Enterobacteriaceae* from Beef Cattle in the North West Province, South Africa: A Critical Assessment of Their Possible Public Health Implications

**DOI:** 10.3390/antibiotics9110820

**Published:** 2020-11-17

**Authors:** Lungisile Tshitshi, Madira Coutlyne Manganyi, Peter Kotsoana Montso, Moses Mbewe, Collins Njie Ateba

**Affiliations:** 1Antimicrobial Resistance and Phage Biocontrol Research Group, Department of Microbiology, School of Biological Sciences, Faculty of Natural and Agricultural Sciences, North West University, Private Bag X2046, Mmabatho 2735, South Africa; Lungisile.Tshitshi@ump.ac.za; 2Faculty of Agriculture and Natural Sciences, University of Mpumalanga, Private Bag X11283, Mbombela 1200, South Africa; Moses.Mbewe@ump.ac.za; 3Unit for Environmental Sciences and Management, North-West University, Potchefstroom 2520, South Africa; madira.manganyi@nwu.ac.za; 4Food Security and Safety Niche Area, Faculty of Natural and Agricultural Sciences, North West University, Private Bag X2046, Mmabatho 2735, South Africa; montsokp@gmail.com

**Keywords:** *Enterobacteriaceae*, multidrug resistance, carbapenemase, ESBL, resistance genes, cattle

## Abstract

Carbapenems are considered to be the last resort antibiotics for the treatment of infections caused by extended-spectrum beta-lactamase (ESBL)-producing strains. The purpose of this study was to assess antimicrobial resistance profile of Carbapenem-resistant *Enterobacteriaceae* (CRE) isolated from cattle faeces and determine the presence of carbapenemase and ESBL encoding genes. A total of 233 faecal samples were collected from cattle and analysed for the presence of CRE. The CRE isolates revealed resistance phenotypes against imipenem (42%), ertapenem (35%), doripenem (30%), meropenem (28%), cefotaxime, (59.6%) aztreonam (54.3%) and cefuroxime (47.7%). Multidrug resistance phenotypes ranged from 1.4 to 27% while multi antibiotic resistance (MAR) index value ranged from 0.23 to 0.69, with an average of 0.40. *Escherichia coli* (*E. coli*), *Klebsiella pneumoniae* (*K. pneumoniae*), *Proteus mirabilis* (*P. mirabilis*) and *Salmonella* (34.4, 43.7, 1.3 and 4.6%, respectively) were the most frequented detected species through genus specific PCR analysis. Detection of genes encoding carbapenemase ranged from 3.3% to 35% (*blaKPC*, *blaNDM*, *blaGES*, *blaOXA-48*, *blaVIM* and *blaOXA-23*). Furthermore, CRE isolates harboured ESBL genes (*blaSHV* (33.1%), *blaTEM* (22.5%), *blaCTX-M* (20.5%) and *blaOXA* (11.3%)). In conclusion, these findings indicate that cattle harbour CRE carrying ESBL determinants and thus, proper hygiene measures must be enforced to mitigate the spread of CRE strains to food products.

## 1. Introduction

Carbapenem-resistant *Enterobacteriaceae* (CRE) strains pose a serious threat, especially in public health worldwide [1,2]. These strains cause severe infections such as bloodstream, pneumonia and complicated urinary tract infections in debilitated immunocompromised patients, thus leading to prolonged hospital stay as well as increased healthcare costs and mortality rates [2,3]. According to the Centers for Disease Control and Prevention (CDC), direct healthcare costs associated with antimicrobial resistance infections are estimated at $20 billion per annum in developed countries [4]. Carbapenem-resistant *Enterobacteriaceae* strains have been commonly reported in hospital settings and patients in intensive care units [1,5,6]. Although *Escherichia coli* (*E. coli*) and *Klebsiella pnuemoniae* (*K. pneumoniae*) are the most frequently detected CRE species, other clinical pathogens such *Citrobacter freundii*, *Enterobacter aerogenes* (*E. aerogenes*), *Enterobacter cloacae* (*E. cloacae*), *P. mirabillis*, *Salmonella* and *Serratia marcescens* species have been detected from environmental samples [7,8]. In addition, several studies have reported that these species harbours clinically important carbapenemase encoding genes such as *blaKPC*, *blaVIM*, *blaIMP*, *blaNDM* and *blaOXA-48* [9]. These genes are usually plasmid-borne, thereby accelerating horizontal transfer of resistance determinants between the same and/or different species [10]. Worrisome is the fact that Carbapenem-resistant strains may carry extended-spectrum beta-lactamase (ESBL) resistance genes.

Several studies have detected major ESBL genes (*blaTEM*, *blaSHV*, *blaCTX-M* and *blaOXA*) in CRE strains [11]. Given the fact that carbapenem antibiotics are considered as the last resort for treating infections caused by ESBL-producing strains, presence of carbapenemase and ESBL genes in CRE strains coupled with lack new therapeutic option is cause for concern [9,10]. Against this background, the World Health Organisation (WHO) has classified CRE strains (*Acinetobacter baumannii* (*A. baumannii*)*, E. coli, K. pneumoniae, Pseudomonas aeruginosa* (*P. aeruginosa*) and *Enterobacter* species) as “critical priority pathogens”, which require urgent research and development novel and effective antibiotics [12]. Furthermore, the WHO has published an “action plan”, which encourages all countries to establish their own action plan to curb antibiotic resistance phenomenon [13]. This could be achieved via extensive research as well as surveillance and epidemiological studies both nationally and internationally. Data obtained from such studies may assist to strengthen the existing departmental and national policies to mitigate the spread of antimicrobial resistance pathogens.

Despite the fact that most of the studies conducted on CRE have been confined within hospital environment and admitted patients [5,6,14], there is a new evidence indicating that food producing animals, especially cattle, pigs and poultry, may harbour CRE carrying ESBL genes [15,16,17,18]. Nevertheless, the number of studies investigating cattle as potential reservoir of CRE harbouring ESBL genes is limited. Hence, the current study was undertaken to determine the occurrence of CRE carrying ESBL determinants in cattle.

## 2. Results

### 2.1. Enterobacteriaceae Isolated from Cattle Faeces

Out of the 233 faecal samples collected from four farms, a total of 280 presumptive isolates belonging to the *Enterobacteriaceae* family were obtained. *Enterobacteriaceae* isolates were mostly obtained from Farm_R-B and Farm_N-D (101 and 92, respectively). Only 59 and 28 of the isolates were obtained from Farm_K-A and Farm_L-C, respectively.

### 2.2. Carbapenem Resistance Isolates

All 280 isolates revealed various antimicrobial susceptibility profiles against four carbapenem antimicrobial agents tested; Figure 1. Of these, 69.3% of the isolates showed intermediate or resistance to one or more of the four carbapenem antibiotics. The isolates revealed resistance to imipenem (42%), ertapenem (35%), doripenem (30%) and meropenem (28%), while intermediate resistance was shown by 26, 31, 33 and 41%, respectively. All 194 isolates that exhibited carbapenem resistance phenotypes were positive for the modified Hodge test. A total of 151 isolates were capable of growing on Brilliance™ ESBL agar and produced colonies with different colours (pink, green, colourless and brown halo), indicating that the isolates could be composed of different species such as *E. coli*, *Klebsiella*, *Enterobacter*, *Serratia*, *Citrobacter* and *Salmonella, Proteus*.

### 2.3. Antimicrobial Resistance Profile of CRE

The 151 isolates that were Carbapenem-resistant and also positive for ESBL production revealed various antimicrobial resistance profiles against 13 different antibiotics tested, Figure 2. The isolates were resistant to at least two or more antimicrobial agents tested. Most of the isolates were resistant to cefotaxime, aztreonam and cefuroxime (59.6, 54.3 and 47.7%, respectively). Resistance to other antimicrobial agents (ceftiofur, amoxicillin, piperacillin, ticarcillin, cephalothin, ceftazidime and cefoxitin) ranged from 39.1 to 43.7%. Low resistance (12.6 and 9.3%) was observed for ciprofloxacin and amoxicillin-clavulanate, respectively. Large proportion (93.4%) of CRE isolates were resistant to three or more antimicrobial agents and were defined as multidrug resistant strains. Multidrug resistance phenotypes ranged from 1.4 to 27%, with the most common MDR pattern being observed against four, five and six different antimicrobial agents (27, 27 and 22%, respectively); Figure 3. Multi antibiotic resistance (MAR) index value ranged from 0.23 to 0.69, with an average of 0.40. The MAR indices 0.31, 0.38 and 0.46 were the most frequently observed among CRE isolates, Figure 4 and Appendix A.

### 2.4. Molecular Identification of CRE Species

A total of 151 CRE isolates were confirmed through amplifying the 16S rRNA conserved region. Of these, 84% were confirmed at species level (*E. coli* (34.4%), *K. pneumoniae* (43.7%), *P. mirabilis* (1.3%) and *Salmonella* (4.6%)) through genus-specific PCR analysis. The remaining 16% of the isolates that could not be identified as either one of these four species were classified as unspecified CRE species.

### 2.5. Detection of Genes Encoding Carbapenemases in CRE

All six carbapenemase-encoding genes screened were detected in CRE isolated from cattle faeces. The *blaKPC* (35.8%), *blaNDM* (20.5%) and *blaGES* (17.9%) were the most frequently detected genes, Figure 5. The *blaOXA-48*, *blaVIM* and *blaOXA-23* were detected in low proportions (10.6, 6.6 and 3.3%, respectively). Simultaneous detection of *blaKPC*_*blaOXA-23* (2.6%) *blaKPC*_*blaNDM* (1.3%) and *blaGES*_*blaOXA-48* (1.3%) genes in some of the isolates were observed. In general, large proportion (94.7%) of CRE species carried carbapenem resistance genes. Carbapenemase encoding genes were commonly detected in *E. coli* and *K. pneumoniae* species (96.2 and 92.4%, respectively), while all *Salmonella* and *Proteus mirabilis* species detected in this study possessed CR determinants; Table 1.

Large proportion (42.4 and 27.3%) of *K. pneumoniae* species harboured *blaKPC* and *blaGES*, followed by 34.6 and 32.7% of *E. coli* species carrying *blaKPC* and *blaNDM*, respectively. *Salmonella* species (42.9 and 28.6%) carried *blaNDM* and *blaKPC*, while unspecified CRE species (29.2 and 25.0%) of possessed of *blaVIM* and *blaKPC* and 50% of *P. mirabilis* species harboured either *blaOXA-23* or *blaOXA-48*. Some of *E. coli* species (3.8%) also possessed *blaKPC*_*blaNDM*, while *K. pneumoniae* species (6.1 and 1.5%) harboured *blaKPC*_*blaOXA-23* and *blaGES*_*blaOXA-48*, followed by unspecified CRE species (4.2%) possessing *blaGES*_*blaOXA-48*.

### 2.6. Detection of ESBL Determinants in CRE

All four major ESBL-encoding genes screened were detected in CRE isolates. The *blaSHV* (33.1%), *blaTEM* (22.5%) and *blaCTX-M* (20.5%) were most frequently detected genes while *blaOXA* (11.3%) detection was very low, Figure 6. Concurrent detection by *blaOXA*_*blaCTX-M* (7.3%) and *blaSHV*_*blaTEM* (5.3%) genes in CRE isolates were observed. Generally, 87.4% CRE species harboured ESBL genes. As indicated in Table 2, most of *E. coli* (86.5%) and *K. pneumoniae* (84.8%) and *Salmonella* (85.7%) species were positive for ESBL genes. All *P. mirabilis* and 95.8% unspecified CRE species carried ESBL genes. Large proportion (45.5%) of *K. pneumoniae* species harboured *blaSHV*. *E. coli* species (34.6 and 26.9%) and *Salmonella* species (57.1 and 28.6%) carried *blaCTX-M* and *blaTEM*, respectively). The 33.3% of unspecified CRE species harboured *blaSHV* while 50% of *P. mirabilis* species carried either *blaTEM* or *blaOXA*. Some *E. coli* (9.6 and 3.8%) and *K. pneumoniae* species (3.0 and 12.1%) possessed *blaSHV*_*blaTEM* and *blaOXA*_*blaCTX-M*, respectively. *Salmonella* (14.3%) and *P. mirabilis* species (9.1%) possessed *blaOXA*_*blaCTX-M*, while unspecified CRE species (4.2%) harboured *blaSHV*_*blaTEM*.

## 3. Discussion

This study reports the occurrence of Carbapenem-resistant *Enterobacteriaceae* from cattle faeces obtained from different commercial farms in the North West province, South Africa. Overall, 280 *Enterobacteriaceae* were successfully isolated. Notably, *Enterobacteriaceae* isolates revealed intermediate or resistance to carbapenem antibiotics (imipenem, ertapenem, doripenem and meropenem) tested. Moreover, resistance to carbapenem antibiotics ranged from 28 to 42%. High resistance was observed against imipenem (42%). This finding was lower than 98% imipenem resistance reported in the previous study [19]. Interestingly, all *Enterobacteriaceae* isolates that revealed intermediate and/or resistant phenotypes against carbapenem antibiotics tested positive for the modified Hodge test and were thus regarded as CRE strains [15,20]. In addition, the isolates were capable of growing on Brilliance™ ESBL media, and this suggests that those isolates carry ESBL determinants [15]. Since carbapenem antibiotics are regarded as being the last resort for the treatment of infections caused by ESBL-producing *Enterobacteriaceae* or multidrug resistant pathogens, resistance to this antibiotic group may exacerbate morbidity and mortality rates in humans [14,21,22,23]. Therefore, monitoring of CRE and the genes associated with CR in food producing animals is essential to determine prevalence of Carbapenem-resistant pathogens in cattle.

Extensive use of antibiotics in agriculture, especially in food producing exacerbate the emergence antibiotic resistant pathogens in food producing animals. Several studies have reported the occurrence of antimicrobial resistant pathogens in foodborne pathogens [24,25,26]. In this study, Carbapenem-resistant *Enterobacteriaceae* isolates revealed various resistance patterns against beta-lactam, 1st, 2nd and 3rd cephalosporins antibiotics. Most of the isolates were highly resistant to cefotaxime, aztreonam and cefuroxime (59.6, 54.3 and 47.7%, respectively). These results corroborate the previous study [26]. In addition, large proportion (93.4%) of CRE isolates revealed MDR phenotypes, which is higher than the 83.78% MRD phenotype previously reported in spinach in South Africa [27]. Moreover, resistance against a maximum of nine antimicrobial agents was observed, suggesting that some of the CRE isolates obtained in this study could possibly be considered as extensively drug resistant (XDR) strains [28]. Interestingly, the MAR index value of 129 MDR strains ranged from 0.23 to 0.69, with an average of 0.40. Although these findings were lower than that of the other study [29], the MAR indices observed in this study were > 0.2. A MAR index of 0.2 or higher indicates high-risk sources of contamination [30,31]. This implies that continuous monitoring of carbapenem resistance in food producing animals, especially cattle is crucial to ensure the safety of food.

Increasing emergence of Carbapenem-resistant species poses a severe threat in public health [32]. Several studies reported have reported the occurrence of CRE in food, animals, water, hospitalised patients and hospitals and/or clinic environments [2,5,33,34]. *E. coli* and *K. pneumoniae* are the most predominant species associated with carbapenem resistance [32]. Overall, four CR species (*E. coli*, *K. pneumoniae*, *Salmonella* and *P. mirabilis*) were identified using genus-specific PCR analysis in this study. Similar to other studies [5,34,35], *K. pneumoniae* (43.7%) and *E. coli* (34.4%) species were the most commonly detected species. Moreover, other CR species, *Salmonella* (4.6%) and *P. mirabilis* (1.3%) were detected in low quantity. However, 16% of CRE isolates could not be identified at the specie level and were classified as ‘unspecified CRE species’.

Carbapenem resistance determinants (*blaKPC*, *blaNDM*, *blaOXA-23*, *blaVIM*, *blaOXA-48* and *blaGES*) have been detected in CRE isolated from different sources such as hospitals, drinking and recreational water, agricultural environments, food producing animals and food products [23]. In this study, 97.4% of CRE isolates possessed carbapenem resistance genes were detected in CRE species. These findings were higher than the 14.3% detection of carbapenem resistance determinants reported in the previous study in from vegetables [33]. This variation could be attributed to different sample sources and isolation methods used per study. Furthermore, high detection of CR genes in this study indicates that the use of carbapenem antibiotics in agriculture may increase the occurrence of CR pathogens in food producing animal, especially cattle. As a result, this may accelerate dissemination of CR pathogens to humans through consumption of contaminated food [22,33,36]. Based on each species, *E. coli* (96.2%), *K. pneumoniae* (92.4%), *Salmonella* (100%), *P. mirabilis* (100%) and unspecified CRE species (95.8%) carried carbapenem resistance determinants. However, these findings were higher than that of the other studies [14,33,37]. However, these findings were higher than those of the other studies [14,33,37]. A possible explanation for this variation could be attributed to differences in the source of samples and geographical location and management practices per farm, which may have different selective pressure for the antimicrobial resistance levels [31]. Most *K. pneumoniae* species and *E. coli* carried *blaKPC*, while *Salmonella* species possessed the *blaNDM* gene. In addition, the *P. mirabilis* species possessed either *blaOXA-23* or *blaOXA-48*, whereas unspecified CRE species possessed of *blaVIM*. Some *E. coli* species possessed a combination of *blaKPC*_*blaNDM*, while *K. pneumoniae* species harboured *blaKPC*_*blaOXA-23* and *blaGES*_*blaOXA-48*, followed by unspecified CRE species possessing *blaGES*_*blaOXA-48*. Given that *E. coli* and *K. pnuemoniae* species cause severe infection in humans, detection of Carbapenem-resistant determinants in these species cannot be overemphasised.

In *Enterobacteriaceae*, ESBL determinants (*blaSHV*, *blaTEM*, *blaCTX-M* and *blaOXA*) are considered as the primary mechanism for mediating beta-lactam antibiotics resistance in *Enterobacteriaceae* [38,39]. Given that extended spectrum cephalosporins antibiotics are used in veterinary medicine, this has resulted in the emergence of ESBL resistance genes in food producing animals, especially cattle [40]. Numerous studies have detected ESBL determinants in *Enterobacteriaceae* isolated from food, cattle and pigs [26,37,41]. Likewise, ESBL-encoding genes (*blaSHV*, *blaTEM*, *blaCTX-M* and *blaOXA*) were detected in this study. Overall, large proportion (87.4%) of CRE isolates harboured ESBL genes. This finding was higher than that of the previous studies, which reported the occurrence of ESBL-producing *Enterobacteriaceae* in cattle and food [26,42,43]. The *blaSHV*, *blaTEM* and *blaCTX-M* were the most frequently (20.5–33.1%) detected genes. However, these findings were lower than those of the previous studies, which reported the prevalence of ESBL-producing strains obtained from hospital environment [27,44,45]. The variation could be attributed to the source of samples, geographical location and the number of isolates analysed per study. Furthermore, *blaOXA* was detected at low rate. Based on each species, large proportion (84.8–100%) of CRE species (*E. coli* (86.5%), *K. pneumoniae* (84.8%), *Salmonella* (85.7%) and *P. mirabilis* (100%) unspecified CRE species (95.8%)) harboured ESBL genes. Large proportions of *K. pneumoniae* (45.5%) and unspecified CRE (33.3%) species harboured *blaSHV*. This finding is consistent with the other study, in which *blaSHV* was predominantly detected in *K. pnuemoniae* [44]. Moreover, *E. coli* (34.6%) and *Salmonella* species (57.1%) possessed *blaCTX-M*, while *P. mirabilis* (50%) species carried either *blaTEM* or *blaOXA*. Notably, some *E. coli* and *K. pneumoniae* species harboured *blaSHV*_*blaTEM*, which is similar to another study from Eastern Cape Province, South Africa [44]. Another ESBL combination observed was *blaOXA*_*blaCTX-M*. These findings suggest that strict measures must be implemented throughout the food chain to mitigate transmission of antimicrobial resistant pathogens to the environment, as well as humans.

## 4. Materials and Methods

### 4.1. Ethical Consideration

Ethical clearance and approval for the study was obtained from the Animal Care Research Ethics Committee (AnimCare REC), of the North-West University, South Africa (Reference number: NWU-00066-15-S9).

### 4.2. Study Area, Sample Collection and Processing

This is a cross-sectional study, and it was conducted from July 2016 to July 1017 in the North-West province, South Africa. A total of 233 faecal samples were collected from four cattle farms (Farm_K-A, Farm_R-B, Farm_L-C and Farm_N-D) located in the Ngaka Modiri Molema district. The selection of the sampling sites was based on the accessibility and the wiliness of the farm owners to participate in the study. The farms had 100 to 400 head of cattle. Sampling was done between July 2016 and August 2017 and all the ethical procedures were followed during handling of the animals. Faecal samples were collected directly from the rectum of individual animals using sterile arm-length gloves and in order to avoid duplication of sampling, the cattle were locked into their respective handling pens. Samples were placed in sterile sample collection bottles, labelled appropriately and immediately transported on ice packs to the Antimicrobial Resistance and Phage Biocontrol Research Group (ARPHBRG) laboratory, North-West University for microbial analysis. For bacteria isolation, 1 g of each sample was dissolved in 2% (*w*/*v*) sterile buffered peptone water (Biolab, Lawrenceville, GA, USA). Aliquot of 5 µL of each sample (mixture) was transferred into 10 mL buffered peptone water. Ten-fold serial dilutions were prepared and aliquots of 100 µL from each dilution was spread-plated on MacConkey agar supplemented with crystal violet and salt (Biolab, Lawrenceville, GA, USA) using a standard procedure. The plates were incubated aerobically at 37 °C for 24 h. The colonies depicting different colours (pale, pink or red) were selected and purified by streaking on MacConkey agar and the plates were at 37 °C for 24 h. Pure colonies were preserved in 20% glycerol and the stock cultures were stored at −80 °C for future use. 

### 4.3. Culture-Based Methods for Identification of Carbapenem Resistance Enterobacteriaceae Colonies

A total of 280 isolates were screened for carbapenem resistance using Kirby–Bauer disc diffusion method [46], the isolates were revived on MacConkey agar and the plates were at 37 °C for 24 h. After incubation, one colony was transferred into 50 mL falcon tubes containing 10 mL nutrient broth. The tubes were incubated at 37 °C for 20 h. The turbidity of the cultures was adjusted to 1 × 108 CFU/mL (equivalent to 0.5 McFarlan standard) using Thermo Spectronic (Model, Helios Epsilon) [Thermo-Fisher Scientific, Waltham, MA, USA]. Aliquot of 100 µL of the culture was spread on Muellerf–Hinton agar (Biolab, Lawrenceville, GA, USA). Four carbapenem antibiotic discs: imipenem (IPM, 10 µg), ertapenem (ETP, 10 µg), meropenem (MEM, 10 µg and doripenem (DOR, 10 µg) (Mast Diagnostics, Randburg, South Africa) were placed on inoculated plates. The plates were incubated at 37 °C for 24 h. The results were interpreted using both Clinical Laboratory Standards Institute and European Committee on Antimicrobial Susceptibility Testing guidelines [20,47]. Colonies of all the isolates showing intermediate or resistant phenotypes to one of the three tested antibiotics were further subjected to Modified Hodge Test to confirm their ability to produce carbapenemase [48]. Carbapenemase-negative *E. coli* (ATCC 25922) and Carbapenemase-positive *K. pneumoniae* (ATCC BAA-1705) were used as quality control.

### 4.4. Phenotypic Screening for Identification of ESBL-Producing Enterobacteriaceae

The isolates that showed intermediate or resistant phenotypes to at least one of the carbapenem antibiotics were screened for the presence of extended spectrum beta-lactamase (ESBL) traits. Briefly, the isolates were culture on chromogenic Brilliance^TM^ ESBL agar (Thermo-Fisher Scientific, Waltham, MA, USA). *E. coli* (ATCC 25922) and *K. pneumoniae* (ATCC 700603) were used as the quality control organisms. Growth on Brilliance^TM^ ESBL agar indicated the ability of the isolates to produce extended spectrum beta-lactamase, and the results were interpreted according to manufacturer’s instructions.

### 4.5. Antimicrobial Susceptibility Test

All the CRE isolates revealing ESBL characteristics on Brilliance^TM^ ESBL agar were subjected to antibiotic sensitivity test to determine their antimicrobial resistance profile according to Kirby–Bauer disc diffusion method [46], following Clinical Laboratory Standard Institute guidelines [20]. The thirteen antibiotics used were: Amoxicillin (A, 25 µg), Amoxicillin-clavulanate (AMC, 30 µg), Aztreonam (ATM, 30 µg), Cefepime (CPM, 30 µg), Cefotaxime (CTX, 30 µg), Cefoxitin (FOX, 30 µg), Ceftazidime (CAZ, 30 µg), Cefuroxime (CXM, 30 µg), Ceftiofur (EFT, 30 µg), Cephalothin (KF, 30 µg), Ciprofloxacin (CIP, 5 µg), Piperacillin (PRL, 30 µg) and Ticarcillin (TC, 75 µg) (Mast Diagnostics, Randburg, South Africa). *E. coli* (ATCC 25922) and *K. pneumoniae* (ATCC 700603) were used as the quality control organisms. The isolates were classified as sensitive, intermediate resistant or resistant based on standard reference values [20]. Any isolate revealing resistance to at least three or more different antibiotics tested was considered multidrug resistant. Multiple antibiotic resistance (MAR) index was determined as the ratio of the number of antibiotics to which CRE isolate showed resistance to the number of antibiotics to which the isolate was exposed [30], using the following formula:(1)MARI=X/Y
where ‘*X*’ is the number of antimicrobial agents which bacteria revealed resistance while ‘*Y*’ is the total number of antimicrobial agents tested.

### 4.6. DNA Extraction from CRE Isolates

Genomic DNA was extracted from all CRE/ESBL-producing isolates using the Zymo Research Genomic DNA^TM^–Tissue MiniPrep Kit (Zymo Research Corp, Irvine, CA, USA) according to the manufacturer’s instructions. The quality and quantity of DNA was determined using Nanodrop^TM^-Lite spectrophotometer (Thermo Scientific, Walton, MA, USA). Pure DNA samples were stored at −20 °C for future use.

### 4.7. Genus-Specific Identification of CRE Isolates

Bacterial universal primer was used to amplify bacterial 16S rRNA gene fragments from the genomic DNA. The identities of the isolates were confirmed by genus-specific PCR, targeting the *uidA*, *ntrA*, *tuf* and *invA* genes specific for *E. coli*, *K. pneumoniae*, *P. mirabilis* and *Salmonella* species, respectively. Details of oligonucleotide primer sequence and PCR conditions are listed in Table 3. PCR reactions were prepared in standard 25 µL (comprising 12.5 µL 2 × DreamTaq Green Master Mix, 0.25 µL of each oligonucleotides primer, 11 µL RNase-DNase free water and 1 µL DNA template). A no DNA template (nuclease-free water) reaction tube served as a negative control while a DNA sample from *E. coli* (ATCC 25922), *K. pneumoniae* (ATCC 700603), *S. enterica* (ATCC 14028 and 12325) were used as quality control organisms. All the PCR reagents were New England Biolabs (Ipswich, MA, USA, supplied by Inqaba, Pretoria, South Africa) products supplied by Inqaba Biotechnical Industry Ltd., Pretoria, South Africa. Amplifications were performed using a DNA thermal cycler (model- Bio-Rad C1000 Touch™ Thermal Cycler). All PCR products were resolved by agarose gel electrophoresis and the rest were stored at 4 °C.

### 4.8. Detection of Carbapenemase-Encoding Genes Using Multiplex PCR

All confirmed CRE isolates were subjected to Multiplex PCR analysis for detection of carbapenemase resistance genes (*blaNDM*, *blaKPC*, *blaVIM*, *blaOXA-23*, *blaOXA-48* and *blaGES*). The oligonucleotide primer sequence and PCR conditions are listed in Table 4. PCR reactions were prepared in standard 25 µL (comprising of 12.5µL 2 × DreamTaq Green Master Mix, 0.25 µL of each oligonucleotides primer, 11 µL RNase-DNase free water and 1 µL DNA template). A non-DNA template (nuclease-free water) reaction tube served as a negative control. DNA sample from *K. pneumoniae* (ATCC BAA-1705), *S. enterica* (ATCC 14028 and 12325) was used as quality control. All the PCR reagents were New England Biolabs (Ipswitch, MA, USA supplied by Inqaba, Pretoria, South Africa) products supplied by Inqaba Biotechnical Industry Ltd., Pretoria, South Africa. Amplifications were performed using a DNA thermal cycler (model-Bio-Rad C1000 Touch™ Thermal Cycler). All PCR products were resolved by agarose gel electrophoresis and the rest were stored at 4 °C.

### 4.9. Detection of Extended Spectrum Beta-Lactamase-Encoding Genes in CRE

The CRE isolates were further screened for the presence of the major ESBL genes (*blaCTX*-M, *blaOXA*, *blaSHV* and *blaTEM*). Details of oligonucleotide primer sequences and PCR conditions are listed in Table 4. PCR reactions were prepared in standard 25 µL (comprising of 12.5 µL 2 × DreamTaq Green Master Mix, 0.25 µL of each oligonucleotides primer, 11 µL RNase-DNase free water and 1 µL DNA template). A no DNA template (nuclease-free water) reaction tube served as a negative control. All the PCR reagents were New England Biolabs (Ipswitch, MA, USA supplied by Inqaba, Pretoria, South Africa) products supplied by Inqaba Biotechnical Industry Ltd., Pretoria, South Africa. Amplifications were performed using a DNA thermal cycler (model: Bio-Rad C1000 Touch™ Thermal Cycler). All PCR products were resolved by agarose gel electrophoresis and the rest were stored at 4 °C.

### 4.10. Agarose Gel Electrophoresis and Visualization

A 2% (*w*/*v*) agarose gel containing 0.1 µg/mL Ethidium bromide was used to separate all PCR products by electrophoresis. A 1 Kb or 100 bp DNA molecular weight marker (Fermentas, Foster City, CA, USA) was included in each gel. A horizontal Pharmacia Biotech equipment system (Model Hoefer HE 99X, Amersham Pharmacia Biotech, Stockholm, Sweden) was used to perform the electrophoresis, each cycle ran for 1 h at 80V. Visualization and image capturing was performed using a ChemiDoc Imaging System (Bio-Rad ChemiDoc^TM^ MP Imaging System, Hercules, CA, USA).

### 4.11. Statistical Analysis

The data generated in this study was entered into an Excel spreadsheet. Descriptive analysis was conducted using SAS, 2010 (v 9.4, SAS Institute, Cary, NC, USA). The proportions of positive for *E. coli* O177 serotype, antimicrobial resistance and virulence genes across the farms were determined. The analysed data were used to draw tables and bar graphs.

## 5. Conclusions

To the best of our knowledge, this is the first study reporting the occurrence of CR- and ESBL-producing *Enterobacteriaceae* in South African beef cattle. The results contained herein revealed high MDR phenotypes among CR- and ESBL-producing *Enterobacteriaceae*. Furthermore, the most frequently detected species were *E. coli* and *K. pneumoniae*. Notably, high detection of carbapenemase and/or ESBL resistance determinants in CRE was alarming given the significant importance of carbapenem antibiotics in public health. Therefore, stringent hygiene measures must be enforced to mitigate the spread and transmission of CRE- and ESBL-producing pathogens in beef farms.

## Figures and Tables

**Figure 1 antibiotics-09-00820-f001:**
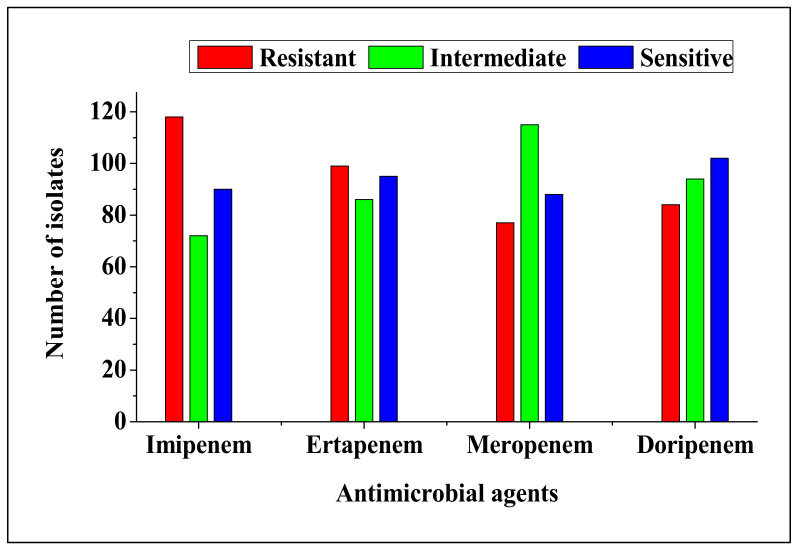
Antimicrobial susceptibility profile of *Enterobacteriaceae* isolated from cattle faeces.

**Figure 2 antibiotics-09-00820-f002:**
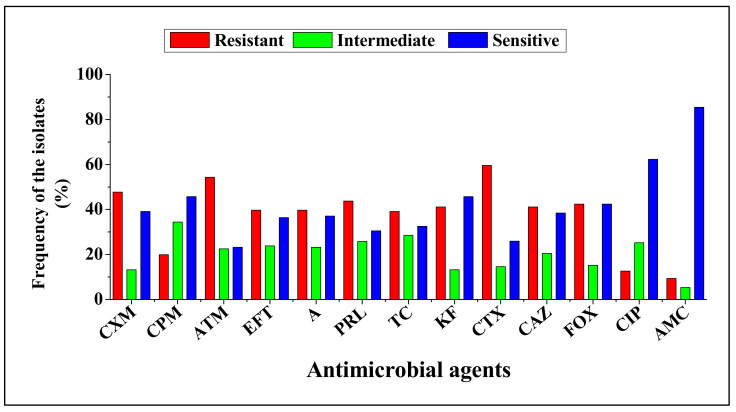
Antimicrobial susceptibility pattern of carbapenem-resistant *Enterobacteriaceae*. A = Amoxicillin, AMC = Amoxicillin-clavulanate, ATM = Aztreonam, CPM = Cefepime, CTX = Cefotaxime, FOX = Cefoxitin, CAZ = Ceftazidime, CXM = Cefuroxime, EFT = Ceftiofur, KF = Cephalothin, CIP = Ciprofloxacin, PRL = Piperacillin and TC = Ticarcillin.

**Figure 3 antibiotics-09-00820-f003:**
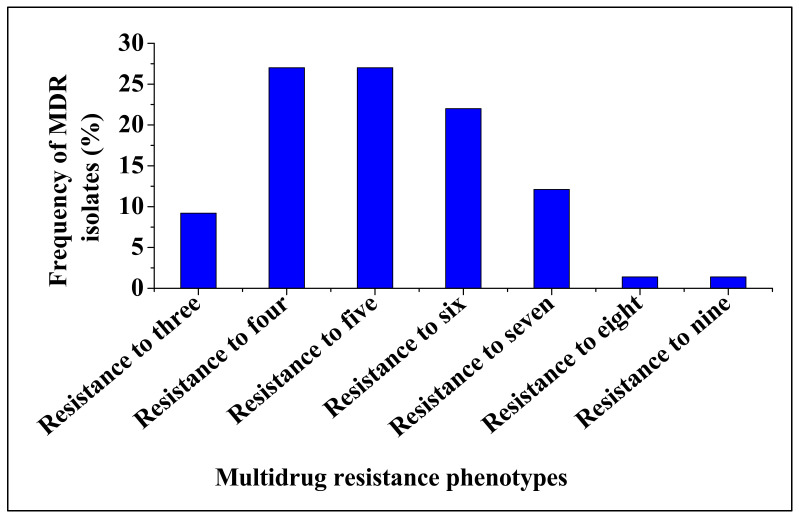
Frequency of multidrug resistance pattern in CRE isolated from cattle faeces.

**Figure 4 antibiotics-09-00820-f004:**
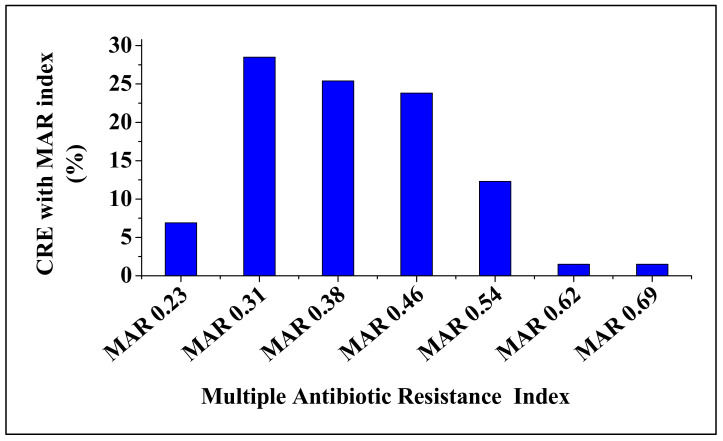
Mean frequency of multiple antibiotic resistance profiles of CRE strains.

**Figure 5 antibiotics-09-00820-f005:**
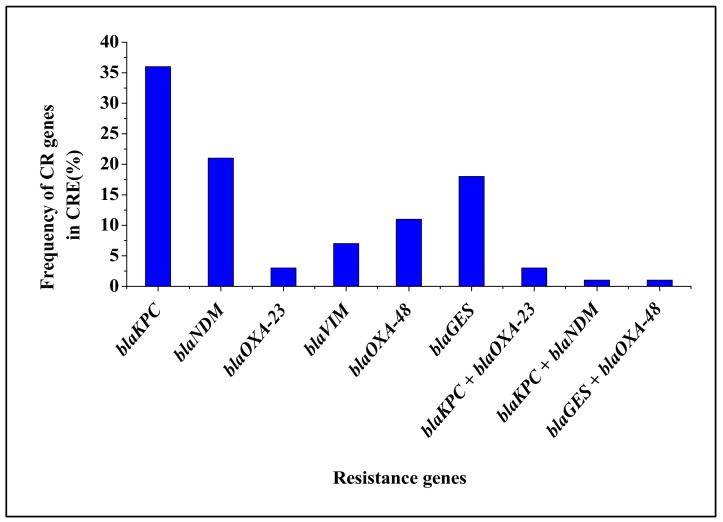
Number of carbapenemase-encoding genes detected in CRE isolated from cattle faeces.

**Figure 6 antibiotics-09-00820-f006:**
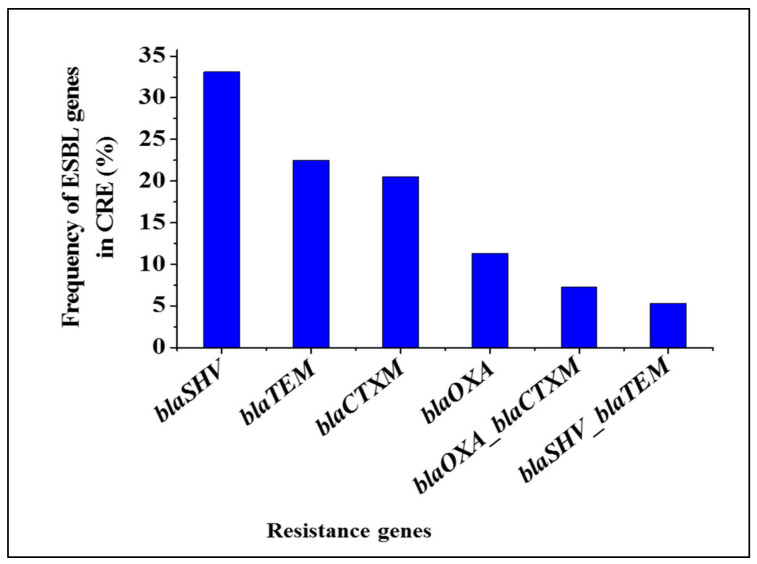
Frequency of ESBL-encoding genes detected in CRE isolated from cattle faeces.

**Table 1 antibiotics-09-00820-t001:** Proportion of CRE species harbouring carbapenemase genes.

CRE Species	No. of Isolates	Carbapenemase Encoding Genes (%)								
		*bla_KPC_*	*bla_NDM_*	*bla_OXA-23_*	*bla_VIM_*	*bla_OXA-48_*	*bla_GES_*	*bla_KPC__bla_OXA-23_*	*bla_KPC_*_*bla_NDM_*	*bla_GES_*_*bla_OXA-48_*
*E. coli*	52	34.6	32.7	3.8	1.9	13.5	9.6	0.0	3.8	0.0
*K. pneumoniae*	66	42.4	12.1	1.5	3.0	6.1	27.3	6.1	0.0	1.5
*P. mirabilis*	2	0.0	0.0	50.0	0.0	50.0	0.0	0.0	0.0	0.0
*Salmonella* species	7	28.6	42.9	14.3	0.0	14.3	0.0	0.0	0.0	0.0
Unspecified CRE species	24	25.0	12.5	0.0	29.2	12.5	16.7	0.0	0.0	4.2
Total	151	35.8	20.5	3.3	6.6	10.6	17.9	2.6	1.3	1.3

**Table 2 antibiotics-09-00820-t002:** Proportion of CRE species carrying ESBL genes.

CRE Species	No. of Isolates	ESBL Encoding Genes (%)					
		*bla_SHV_*	*bla_TEM_*	*bla_CTX-M_*	*bla_OXA_*	*bla_OXA_*_*bla*_CTX-M_	*bla_SHV__bla_TEM_*
*E. coli*	52	23.1	26.9	34.6	1.9	3.8	9.6
*K. pneumoniae*	66	45.5	16.7	9.1	13.6	12.1	3.0
*P. mirabilis*	2	0.0	50.0	0.0	50.0	9.1	0.0
*Salmonella* species	7	0.0	28.6	57.1	0.0	14.3	0.0
Unspecified CRE species	24	33.3	25.0	12.5	25.0	0.0	4.2
Total	151	33.1	22.5	20.5	11.3	7.3	5.3

**Table 3 antibiotics-09-00820-t003:** List of oligonucleotide primer sequences and PCR conditions used in this study.

Primers	Oligonucleotide Sequence (5′–3′)	Genes	Amplicon Size (bp)	Annealing Tm (°C)	References
**16S rRNA**					
27F	AGAGTTTGATCATGGCTCAG	16S rRNA	1420	55	[49]
1492R	GGTACCTTGTTACGACTT				
**Genus Specific Genes**					
ntrA-F	CATCTCGATCTGCTGGCCAA	*ntrA*	90	52	[50]
ntrA-R	GCGCGGATCCAGCGATTGGA				
uidA-F	CTGGTATCAGCGCGAAGTC	*uidA*	556	52	
uidA-R	AGCGGGTAGATATCACACTC				
Tuf-F	TCTACTTCACACGTAG	*tuf*	240	58	[51]
Tuf-R	TTCTAACAGCTCTTCA				
invA-F	GTGAAATTATCGCCACGTGGCAA	*invA*	284	64	[52]
invA-R	TCATCGCACCGTCAAAGGAACC				

**Table 4 antibiotics-09-00820-t004:** List of oligonucleotide primer sequences and PCR conditions used in this study.

Primers	Oligonucleotide Sequence (5′–3′)	Genes	Amplicon Size (bp)	Annealing Tm (°C)	References
**CRE Genes**					
KPC-F	CGTCTAGTTCTGCTGTCTTG	*bla* _KPC_	798	52	[53]
KPC-R	CTTGTCATCCTTGTTAGGCG				
NDM-F	GGTTTGGCGATCTGGTTTTC	*bla* _NDM_	621		
NDM-R	CGGAATGGCTCATCACGATC				
OXA-23-F	ATGAGTTATCTATTTTTGTC	*bla* _OXA-23_	501		
OXA-23-R	TGTCAAGCTCTTAAATAATA				
GES-C-F	GTTTTGCAATGTGCTCAACG	*bla* _GES_	371		
GES-D-R	TGCCATAGCAATAGGCGTAG				
VIM-F	GATGGTGTTTGGTCGCATA	*bla* _VIM_	390		
VIM-R	CGAATGCGCAGCACCAG				
OXA-48-F	TTCGGCCACGGAGCAAATCAG	*bla_OXA-48_*	438		
OXA-48-R	GATGTGGGCATATCCATATTCATCGCA				
**ESBL Genes**					
blaTEM-F	AAACGCTGGTGAAAGTA	*bla_TEM_*	822	45	[54]
blaTEM-R	AGCGATCTGTCTAT				
blaSHV-F	ATGCGTTATATCGCCTGTG	*bla* _SHV_	753		
blaSHV-R	TGCTTTGTTATTCGGGCCAA				
blaCTX-M-F	CGCTTTGCGATGTGCAG	*bla* _CTX-M_	550		
blaCTX-M-R	ACCGCGATATCGTTGGT				
blaOXA-F	ATATCTCTACTGTTGCATCTCC	*bla* _OXA_	619		
blaOXA-R	AAACCCTTCAAACCATCC				

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
