# Peer review of "Extended Spectrum Beta-Lactamase-Resistant Determinants among Carbapenem-Resistant Enterobacteriaceae from Beef Cattle in the North West Province, South Africa: A Critical Assessment of Their Possible Public Health Implications"

_antibiotics, 2020, doi:10.3390/antibiotics9110820_

Round 1

Reviewer 1 Report

The paper entitled “Extended spectrum beta-lactamase resistant determinants among carbapenem resistant Enterobacteriaceae from beef cattle in the North West Province, South Africa: A critical assessment of their possible public health implications” is well designed, executed and written. The paper shows how disseminated the antibiotic resistances are. And although the number of farms is not representative of the country, the information that the paper provides is very important from a public health point of view.

Specific comments:

Line 23: extended-spectrum beta-lactamase instead of extended beta-lactamase

The first time that a bacterium is mentioned, the full name must be written, and after that you could use the shot name. Examples:

Line 48: Escherichia coli, and then E. coli. Or even Escherichia (E.) coli

Line 60: Acinetobacter baumannii instead of A. baumannii

Line 55: add extended-spectrum beta-lactamase (ESBL), because this is the first time in the text (apart from the abstract) that appear this acronym.

Line 61: P. aeruginosa and Enterobacter species instead of P. aeruginosa and Enterobacter species

Line 76: Three or four farms? Farm_R-B, Farm_N-D, Farm_K-A and Farm_L-C

Lines 80-81: Figure 1 gives redundant information as in the text. It could be deleted.

Lines 88-89: “A total of 151 isolates were capable of 89 growing on Brilliance™ ESBL agar and produced colonies with different colours.” More extended information is needed explaining the results.

Line 119:  Salmonella is a genus not a specie, rephrase the sentence.

Author Response

Response to Reviewer 1 Comments

Point 1: Line 23: extended-spectrum beta-lactamase instead of extended beta-lactamase

Response 1: Line 23: – “extended beta-lactamase” has been corrected to “extended-spectrum beta-lactamase”.

Point 2: The first time that a bacterium is mentioned, the full name must be written, and after that you could use the shot name. Examples:

Line 48: Escherichia coli, and then E. coli. Or even Escherichia (E.) coli.

Line 60: Acinetobacter baumannii instead of A. baumannii

Response 2: Full names of the bacteria have been given in their first motioned as indicated within the text.

Point 3: Line 61: P. aeruginosa and Enterobacter species instead

Response 3: P. aeruginosa and Enterobacter species have been italicized.

Point 4: Line 76: Three or four farms? Farm_R-B, Farm_N-D, Farm_K-A and Farm_L-C

Response 4:  “Four farms”

Point 5: Lines 80-81: Figure 1 gives redundant information as in the text. It could be deleted.

Response 5: Figure 4 has been deleted

Point 6: Lines 88-89: “A total of 151 isolates were capable of 89 growing on Brilliance™ ESBL agar and produced colonies with different colours.” More extended information is needed explaining the results.

Response 6: More information has been added and the sentence read as “A total of 151 isolates were capable of growing on Brilliance™ ESBL agar and produced colonies with different colours (pink, green, colourless and brown halo), indicating that the isolates could be composed of different species such as E. coli, Klebsiella, Enterobacter, Serratia, Citrobacter and Salmonella, Proteus”.

Point 7: Line 119: Salmonella is a genus not a specie, rephrase the sentence.

Response 7: The sentence has been rephrased and read as “Of these, 84% were confirmed at specie level [E. coli (34.4%), K. pneumoniae (43.7%), P. mirabilis (1.3%) and Salmonella (4.6%)] through genus specific PCR analysis”

Reviewer 2 Report

The authors described the cattle-derived Enterobactericeae strains resistant to carbapenems from South Africa. The problem is very important to public health.
In the introduction - line 61 - P. aeruginosa and Enterobacter - should be in italics;
The methodology was properly selected. The presented results are adequate to the assumed goal of the work. 
A few comments on the interpretation of the results:
Figures 3 and 4 - the description of the Y axis is confusing, it has to be decided whether the number of isolates (number) or their frequency (%) are presented.
Figures 5,6 and 7 - the description in the graph field under the X axis is a repeat of the caption, it should be changed.

In the discussion - lines 171-173 - The meaning of the sentence is unclear. Screening allows the assessment of the scale of the problem, but is not a tool for limiting the phenomenon. - it should be corrected
Lines 203-206 and 209-212 - repeating the content of the results. The discussion cannot duplicate the content of the results. The results should be discussed in terms of the causes that cause drug resistance in the strains, such as whether cattle are given antibiotics as prophylaxis. It is known that animal husbandry uses many antibiotics. Are the animals given antibiotics for various infections, or is antibiotic therapy rational - too little or too much medication. The conditions of animal husbandry also play a large role, whether the animals stay in barns or pastures, or what quality is the water for the animals.
The conclusions should emphasize the risk of transmission of drug-resistant strains to humans via the food chain as a real problem for public health.

Author Response

Response to Reviewer 2 Comments

Point 1: line 61 - P. aeruginosa and Enterobacter- should be in italics. 

Response 1: line 63: “P. aeruginosa and Enterobacter” have been italicized.

Point 2: Figures 3 and 4 - the description of the Y axis is confusing, it has to be decided whether the number of isolates (number) or their frequency (%) are presented

Response 2: The description of the Y axis in Figures 3: Number of isolates (%) and Figure 4: Number of isolates (%) have been changed to “Figure 2: Frequency of the isolates (%) and Figure 3: Frequency of MDR isolates (%).

Point 3: Figures 5, 6 and 7 - the description in the graph field under the X axis is a repeat of the caption, it should be changed.

Response 3: Figures 5, 6 and 7and their description in the graph field under the X axis have been changed to Figures 4, 5 and 6. The caption for “Figure: 4 has been changed to “Mean frequency of multiple antibiotic resistance profiles of CRE strains”. The description in the graph field under the X axis have been changed to Resistance genes (Figure 5 and 6).

Point 4: lines 171-173 - The meaning of the sentence is unclear

Response 4: Line s172-174: The sentence has been revised and read thus “Since carbapenem antibiotics are regarded as the last resort for the treatment of infections caused by ESBL-producing Enterobacteriaceae or multidrug resistant pathogens, resistance to this antibiotic group may exacerbate morbidity and mortality rates in humans”.

Point 5: Screening allows the assessment of the scale of the problem, but is not a tool for limiting the phenomenon. - it should be corrected.

Response 5: Lines 175-176: The sentences has been corrected and read as: “Therefore, monitoring of CRE and the genes associated with CR in food producing animals is essential to determine prevalence of carbapenem resistance in cattle.”

Point 6: Lines 203-206 and 209-212 - repeating the content of the results. The results should be discussed in terms of the causes that cause drug resistance in the strains.

Response 6: Lines 204-2011: The section has been revised and read thus “In this study, 97.4% of CRE isolates possessed carbapenem resistance genes were detected in CRE species.  These findings were higher than the 14.3% detection of carbapenem resistance determinants reported in the previous study in from vegetables [33]. This variation could be attributed to different sample sources and isolation methods. Furthermore, high detection of CR genes in this study indicates that the use of carbapenem antibiotics in agriculture may increase the occurrence of CR pathogens in food producing animal, especially cattle. As a result, this precipitate the dissemination of CR pathogens to humans through consumption of contaminated food”. However, lines 2011-212- could not be changed because they detail the detection carbapenem resistance genes detected in each specie. Furthermore, this makes it easier to compare the results contained herein with other studies. Therefore, the authors request that the line should remain as is.

Round 2

Reviewer 2 Report

All my comments have been taken into account by the authors, and the scientific value of the manuscript has been improved.